# Metabolome and Transcriptome Analyses Unravel the Molecular Regulatory Mechanisms Involved in Photosynthesis of *Cyclocarya paliurus* under Salt Stress

**DOI:** 10.3390/ijms23031161

**Published:** 2022-01-21

**Authors:** Lei Zhang, Zijie Zhang, Shengzuo Fang, Yang Liu, Xulan Shang

**Affiliations:** 1College of Forestry, Nanjing Forestry University, Nanjing 210037, China; zhanglei321@njfu.edu.cn (L.Z.); iszhangzj@sina.com (Z.Z.); lyang_188@sina.com (Y.L.); shangxulan@njfu.edu.cn (X.S.); 2Co-Innovation Centre for Sustainable Forestry in Southern China, Nanjing Forestry University, Nanjing 210037, China

**Keywords:** wheel wingnut, photosynthetic system, carbon metabolism, regulatory network, transcription activator, abiotic stress

## Abstract

Photosynthesis is the primary life process in nature, and how to improve photosynthetic capacity under abiotic stresses is crucial to carbon fixation and plant productivity. As a multi-functional tree species, the leaves of *Cyclocarya paliurus* possess antihypertensive and hypoglycemic activities. However, the regulatory mechanism involved in the photosynthetic process of *C. paliurus* exposed to salinity has not yet been elucidated. In this study, the photosynthetic characteristics of *C. paliurus* seedlings, such as photosynthetic rate (*P_n_*), stomatal conductance (*G_s_*), and electron transfer rate (ETR), were investigated under different salt concentrations, while the metabolome and transcriptome analyses were conducted to unravel its molecular regulatory mechanisms. Salt stress not only significantly affected photosynthetic characteristics of *C. paliurus* seedlings, but also severely modified the abundance of metabolites (such as fumaric acid, sedoheptulose-7-phosphate, d-fructose-1,6-bisphosphate, and 3-phospho-d-glyceroyl phosphate) involved in central carbon metabolism, and the expression of photosynthetic genes. Through the co-expression network analysis, a total of 27 transcription factors (including *ERF*s, *IDD*, *DOF*, *MYB*, *RAP*) were identified to regulate photosynthetic genes under salt stress. Our findings preliminarily clarify the molecular regulatory network involved in the photosynthetic process of *C. paliurus* under salt stress and would drive progress in improving the photosynthetic capacity and productivity of *C. paliurus* by molecular technology.

## 1. Introduction

Photosynthesis refers to the process of converting light energy into chemical energy and its storage as organic matter (e.g., carbohydrates) for supporting plants’ activities, and is considered as one of the most basic processes supporting life on earth. Improving plant photosynthetic efficiency can ameliorate many of mankind’s major problems, such as food security, resources, and environmental sustainability [1]. The absorption and transformation of light energy require coordination and participation of about 3,000 different photosynthetic protein components, most of which are encoded by nuclear genes, and only a few are encoded within the chloroplast genome [2]. Transcription factors (TFs) and downstream transduction signals of photoreceptors coordinate and control gene transcription and expression between these two genomes (i.e., nuclear and chloroplast), subsequently affecting photosynthesis protein activity [3]. For example, the analysis of genomic binding sites showed that the TF HY5 could promote the enrichment of the direct target genes involved in the photosynthesis process [4], and it had been proven to be closely related to photosynthetic activity regulation [5,6]. Additionally, the signal transmission between chloroplast and nucleus is also mediated by metabolite accumulations [7,8]. For instance, 3′-phosphoadenosine-5′-phosphate could move between the two organelles and regulate nuclear genes expression [9]. Glucose signaling has also been proven to be mediated by hexokinase to alter the expression of photosynthesis genes and their regulators [10,11]. Thus, research on metabolism and transcription levels is a highly effective approach to improving photosynthesis efficiency [12].

Soil salinization, a process of land degradation, limits plant growth and global food production at large [13]. Salinity causes various physiological and biochemical reactions in plants, among which photosynthesis is the main target of the salt effect. A set of physiological activities during photosynthesis, such as stomatal closure, electron transport, leaf photochemistry, and carbon metabolism, are affected by salinity-caused stress [14]. Many studies have identified that abiotic factors such as salinity and drought lead to the production of reactive oxygen species (ROS) molecules H_2_O_2_ and O^2−^ at photosystem I, singlet oxygen at photosystem II, and redox variation at plastoquinone [8,15,16]. These signals, together with approaches such as isoprenes, phosphoadenosines, and protein kinases, construct a communication network between chloroplast and nucleus to regulate the expression of photosynthetic genes [17]. However, there is a paucity of information on the TFs mediating these activities, indicating that the coordination mechanism in response to salt stress in the photosynthetic process has not been fully clarified [18].

Wheel wingnut (*Cyclocarya paliurus* (Batal)Iljinskaja), also known as “sweet tea tree”, is a multifunctional tree species and widely distributed in the subtropical region of China [19]. Its leaves contain plenty of flavonoids, triterpenes, and other bioactive ingredients, that are effective in preventing hyperglycemia and diabetes [20]. Not only used as a traditional Chinese medicine formula, *C. paliurus* leaves are also used in tea production [21]. Owing to its commercial values, a huge production of *C. paliurus* leaves is required for tea production and medical use, therefore developing its plantation is the best option for leaf production as well as for protecting its natural forests [22]. However, it is hard to find enough sites with good soil conditions to plant *C. paliurus* due to the limitation of land resources in China, therefore the land with the abiotic stress, such as coastal saline land, would be the potential areas for establishing *C. paliurus* plantations. A few studies have explored the strategies of external ions alleviating sodium toxicity in *C. paliurus* seedlings under salt stress [23,24], whereas the transcriptional regulatory network under salinity still remains unknown in this species.

Recently, increasing biomass and improving tolerance by mediating plant photosynthetic related processes have attracted extensive attention, thus, a systematic understanding of the molecular mechanisms underpinning photosynthesis regulation is strongly required [25]. Here, we integrated *C. paliurus* leaves physiological and omics data to: (1) analyze the variations of photosynthetic characteristics and carbon metabolism under different salt concentrations; (2) identify key stress-response genes and TFs involved in photosynthesis responding to salt conditions; and (3) expound the related transcriptional regulatory networks. The results of this research would be essential for the improvement of yield and photosynthetic carbon assimilation ability of *C. paliurus* leaves under salt stress.

## 2. Results

### 2.1. Variations in Photosynthetic and Fluorescence Characteristics

Salt stress significantly affected the photosynthesis of *C. paliurus* leaves (*p* < 0.05). The leaves withered and gradually changed color to yellow (Figure 1a) and the photosynthetic parameters significantly changed under the different salt treatments (Table 1). At the two sampling times, intercellular CO_2_ concentration (*C_i_*) was lower in the control compared to the salt treatments. Conversely, net photosynthetic rate (*P_n_*), transpiration rate (*T_r_*), and stomatal conductance (*G_s_*) decreased with the increase of salt concentration, and significant differences (*p* < 0.05) were observed under the different treatments (Table 1). Notably, salt treatments caused an appreciable steady decrease of *P_n_* with increasing salt concentrations. For instance, the values of *P_n_* in LS, MS and HS treatments decreased by 55.24, 70.19, and 73.25% at T_2,_ respectively, compared to the CK. Moreover, the value of *P_n_* in each treatment at T_2_ was lower than that at T_1_, and the *P_n_* value in LS and MS decreased the most (Table 1).

Generally, over treatment time, salinity induced a steady reduction of chlorophyll content with chlorophyll a and b contents showing significant differences among the treatments (*p* < 0.05), and these detected differences were more obvious at T_2_ (Figure 1b,c). A similar changing trend was also observed in the electron transfer rate (ETR), maximal quantum yield of PSII (F_v_/F_m_), and photochemical quenching coefficient (qP) (Figure 1d,e,g). At T_1_, ETR and qP showed no significant differences among the salt treatments and were significantly lower than CK. Nevertheless, the non-photochemical quenching coefficient (NPQ) increased with the salinity increase (Figure 1f).

### 2.2. Metabolomic Response to Salt Stress

Total N and C contents in leaves were closely related to light energy utilization efficiency. Salt treatments induced lower contents for total N and total C compared to the CK (Appendix A). At T_1_, and relative to the control, only MS and HS produced significantly lower total C content; while at T_2_, the total C content of CK was higher than all salt treatments. Furthermore, the total C content in CK and LS at T_2_ was significantly lower than that at T_1_. Similarly, salt stress reduced total N content, and significant differences were only observed in LS and MS between two sampling times.

Based on the comparative analysis of metabolome data, 32 photosynthetic-related metabolites were identified, and the majority significantly differed among the salt treatments (Appendix A). To better understand the relationships among the metabolites in central metabolic processes, we constructed a combination diagram of related metabolic pathways (Figure 2a). According to the heatmap of abundance, we found that metabolites in different metabolic pathways showed different accumulation trends under salt stress (Figure 2b). For instance, the abundance of metabolites in the citrate cycle pathway such as isocitric, fumaric, and succinic acids, as well as acetyl-CoA under salt treatments decreased significantly (*p* < 0.05). On the contrary, sedoheptulose-7-phosphate, d-fructose-1,6-bisphosphate, sedoheptulose, and 3-phospho-d-glyceroyl phosphate in the carbon fixation pathway of a photosynthetic organism, whose abundance was observed to increase under MS and HS treatments compared with the control (CK). Moreover, pyruvic acid and phosphoenolpyruvate, which were involved in multiple metabolic pathways, showed lower abundance at T_2_ than T_1_.

### 2.3. Transcriptomic Response to Salt Stress

The Illumina raw sequencing profiles were submitted to the NCBI BioProject database under number PRJNA700136, and approximately 1.0 billion clean reads were identified from the 24 cDNA libraries. According to quantified gene expression files, principal component analysis (PCA) was performed to evaluate the mutual relationship among the 24 samples (Figure 3a). PCA plots for each sampling time showed a clear separation between samples from different treatments, indicating that *C. paliurus* leaves underwent obvious transcriptional reprogramming under salt stress. Comparing the gene expression between the control (CK) and the different salt treatments, the number of differentially expressed genes (DEGs) gradually increased from low to high salt concentration (Figure 3b). Additionally, more DEGs were observed between CK and each salt treatment at T_2_ than T_1_. For instance, the number of DEGs between CK and HS at T_2_ was the largest (8642 up-regulated and 7810 down-regulated genes), yet the number was only 7275 at T1 (3657 up-regulated and 3618 down-regulated genes) (Figure 3c).

### 2.4. Gene Co-Expression Network Analysis

A total of 19 modules with similar gene expression patterns were identified by weight gene co-expression network analysis (Figure 4a), of which, the cyan module contained the most genes (*n* = 6424) and orangered 3 module contained the least genes (*n* = 59) (Figure 4b). According to the KO (KEGG Ontology) enrichment analysis of each module, the DEGs in the photosynthesis pathway (ko00195) were identified to be significantly enriched in the cyan module (Figure 4c). Meanwhile, the result of KEGG enrichment analysis showed DEGs in the cyan module were significantly enriched in the photosynthesis, porphyrin and chlorophyll metabolism, and photosynthesis antenna proteins pathways (Appendix A). Moreover, GO enrichment analysis revealed that DEGs in the cyan module were mainly related to plastid organization and organism membrane organization in the biological process term, disulfide oxidoreductase activity and carboxylic acid transmembrane transporter activity in the molecular function term, and plastid part in the cellular component term (Appendix A).

In order to explore the regulatory network coordinating the expression of photosynthetic-related genes under salt stress, 42 DEGs in the photosynthesis pathway in the cyan module were screened as target genes (Appendix A). Based on the promoter analysis of the target genes, we identified 27 TFs with a potential regulatory relationship (Appendix A). Further, the correlations between photosynthetic system genes and TFs were displayed by a co-expression analysis network (*p* < 0.05, |r| > 0.75) (Figure 5a). According to this network, *PSBR* (Unigene0017274), *PSAD* (Unigene0081034), *PSAF* (Unigene0049595), *PETH* (Unigene0008364), and *PETE* (Unigene0070030) were found to be positively or negatively regulated by multiple TFs (degree ≥ 4). Collectively, photosynthetic genes expression gradually decreased with the increase of salt concentration and treatment time (Figure 5b). Finally, we constructed a schematic diagram of TFs involved in the photosynthetic process to better understand the regulatory relationships in the photosynthesis pathway under salt stress (Figure 6). The biological processes of photosystem Ⅰ and photosystem Ⅱ were regulated by the most TFs, including *ERF*s, *IDD*, *DOF*, *MYB*, *RAP*, *BPC*, *MYR,* and so on as seen in the network (Figure 6).

## 3. Discussion

Salt stress is a major environmental factor that adversely affects tree growth and productivity [13]. A range of intrinsic mechanisms and adaptive strategies of model species to cope with salinity has been found [14], however, the coordination mechanism in response to salt stress in the photosynthetic process has not been fully elucidated. Here, we clarified the molecular regulatory network involved in the photosynthetic process of *C. paliurus* under salt stress and would drive progress in improving the photosynthetic capacity and productivity of *C. paliurus* by molecular technology.

### 3.1. Effects of Salt Stress on Photosynthetic Performance

As a major abiotic factor affecting plant growth, salt stress induces the increase of antioxidant enzyme activity and the content of low-molecular-mass antioxidants [26], disturbs plant water balance, and results in stomatal closure and photosynthetic efficiency reduction [27]. Our study showed a similar result that *G_s_*, *T_r_*, and *P_n_* decreased significantly under salt treatments (Table 1). As reported, stomatal and non-stomatal factors may lead to plant photosynthetic rate decline under stress [28,29]. According to the observed increase of *C_i_* and decrease of *G_s_* with the increase of salt concentration, we inferred that the decline of *P_n_* in *C. paliurus* leaves is mainly caused by non-stomatal factors, where photosynthetic organs were damaged by salt stress, resulting in the reduction of CO_2_ utilization efficiency in photosynthesis. Our result is in agreement with the result reported by Flexas et al. [30] in tobacco plants.

Chlorophyll is the main pigment of plant photosynthesis and plays a central role in the process of light absorption [31,32]. Consistent with studies on tomato [33] and cucumber [34], the chlorophyll a and b contents in *C. paliurus* leaves significantly decreased under salt stress, and the downward trend was more serious with the extension of treatment time (Figure 1b,c). The possible reason is that under continuous salt stress, the degradation rate of chlorophyll on the thylakoid membrane is faster than that of synthesis, resulting in the decrease of chlorophyll content and the exposure of carotenoids and anthocyanins. Meanwhile, the leaves from high salt concentration showed an obvious color change phenomenon from green to yellow in the present experiment (Figure 1a).

In the photosynthetic system, after light energy is absorbed and transformed, most of the energy turns to photochemical reaction, electron transfer, photosynthetic phosphorylation and CO_2_ fixation, part is dissipated in the form of heat, and a small part is emitted in the form of fluorescence [35]. In our study, a significant decrease of F_v_/F_m_ (Figure 1e) and qP (Figure 1g) was observed under the MS and HS treatments, indicating the impairment of photochemical conversion efficiency in PSII protein complexes at middle and high salinity levels occurred [36,37]. However, NPQ, as the common index to reflect the level of heat dissipation dependent on xanthophyll cycle in PSII complexes [38,39,40], showed an increasing tendency under salt stress (Figure 1f), suggesting that the thermal dissipation was enhanced to avoid photosynthetic structure damage caused by salinity. Additionally, no significant difference in ETR was detected between salt treatments and CK at T_2_ (Figure 1d), suggesting that the excess electrons in the photosynthetic electron transport chain might be assigned to potential electron warehouses such as Mehler reaction and cyclic electron flow under the adverse conditions [41,42].

### 3.2. Effects of Salt Stress on Carbon Metabolism

Salinity affects the photosynthetic efficiency of plants by intervening in chlorophyll biosynthesis, gas exchange, electron transfer mechanisms, photochemical system, metabolic strategies, and so on [25]. The metabolic activity of cells in leaves mainly depends on the Calvin cycle process, which reduces CO_2_ by consuming adenosine triphosphate (ATP) and nicotinamide adenine dinucleotide phosphate (NADPH) to complete carbon fixation [43]. Interestingly, the total carbon content in the leaves of CK was the highest (Appendix A), whereas the maximum abundance of 3-phospho-d-glyceroyl phosphate, d-fructose-1,6-phosphate, and sedoheptulose-1,7-bisphosphate involved in the carbon fixation pathway were observed under MS and HS treatments (Figure 2b). We speculated that the synergistic effect of energy distribution and carbon metabolism led to the increase in abundance of these metabolites under salt stress. For example, higher intercellular CO_2_ concentration can affect photorespiration efficiency [44], thus limiting the production of phosphoglycolate, which could inhibit sedoheptulose-1,7-bisphosphatase and phosphofructokinase activities [45,46,47]. Glycolysis, the citrate cycle, and the pentose phosphate pathway, as the earliest discovered metabolic pathways, play an important role in maintaining intracellular carbon balance, regulating nucleotide and amino acid biosynthesis, and keeping redox balance under various stress situations [48]. In the present study, the abundance of fumaric acid, l-malic acid, and succinic acid significantly decreased under salt stress, which is consistent with the variation tendency of the photosynthetic rate (Appendix A), indicating that these metabolites might be involved in osmotic regulation, stomatal closure, and the maintenance of cell homeostasis, in agreement with the results from *Arabidopsis thaliana* [49,50], *Vicia faba* and *Commelina communis* [51]. Moreover, several studies indicated that the increase of redox level would promote citric acid to flow into the amino acid pathway [52,53], and the excessive accumulation of amino acids disturbs the charge balance in cells and affects the normal physiological function of chloroplasts [54,55]. Conformably, citric acid accumulation was induced by salt stress in our study (Figure 2b), which may interfere with the physiological processes related to photosynthesis. Therefore, we infer that salt stress induced the redistribution of photosynthetic products in *C. paliurus* leaves on metabolic networks such as sucrose, starch, and amino acid synthesis.

### 3.3. Transcriptional Regulation in Photosynthetic System to Salt Stress

It is known that the stress responses of plants to salinity are accompanied by alterations of expression patterns of a large number of genes, and these alterations are controlled by stress intensity, duration, and other factors [56]. Consistently, *C. paliurus* leaves underwent severe transcriptional reprogramming under salt treatment (Figure 3a), and more DEGs were identified with increasing salt concentrations and extending treatment periods (Figure 3c). Photosynthesis, involving a multitude of coding genes and functional proteins to complete light energy absorption and transformation, is one of the primary processes affected by environmental stress [57]. It has been reported that plants respond to external stress through rapid gene expression changes, and salinity can induce the downregulation of related photosynthetic genes expression [56,58]. Consistent with previous studies, the expression of a large proportion of photosynthetic genes was significantly inhibited under salt stress (Figure 4c), resulting in a downward trend as a whole (Figure 5b). Notably, under stress conditions, specific transcription factors (TFs) regulate the photosynthetic genes expression by binding cis-acting elements in the promoters of these target genes [59]. Hence, a gene co-expression network was constructed in the present study to illustrate the regulatory relationship between TFs and target genes in the *C. paliurus* leaves (Figure 5a).

In the regulatory network, multiple ERFs were found to be involved in regulating the stress responses of genes encoding photosystem I (PS I), the ATP synthase complex, and ferredoxin–NADP reductase (FNR) (Figure 6). The AP2/ERF family is generally considered to be a large transcription factor family, including several subfamilies such as RAP, ERF, and DREB, which could activate the expression of stress response genes in abiotic disturbances [60,61,62]; while DREB family transcription factors had been proven to activate the transcription of related responsive genes under salt stress by binding to dehydration-responsive element (TACCGACAT) in their promoters [63]. In our study, *DREB* (Unigene0032570) was identified to positively regulate *PSBR* gene expression. Interestingly, two *NAC*s (Unigene0007735 and Unigene0048500) were significantly and negatively correlated (*p* < 0.05, |r| > 0.75) with photosynthetic genes expression such as *PETE*, *PSAG*, and *PSAH* (Figure 5a). Previous studies have shown that the NAC transcription factor was usually induced in guard cells in response to salt stress and was related to stomatal closure [64]. Furthermore, we found PSAN to be positively regulated by *IDD* (Unigene0013496) and PSAF was negatively regulated by *DOF* (Unigene0050441). IDD proteins could function as inhibitory factors of stomatal initiation [65], and DOF played a crucial role in gene expression related to the light-dependent reaction of photosynthesis [66]. In this study, we performed transcription level analysis to explain the complex network relationship between stress-response genes and transcription factors related to the photosystem, but further exploration and verification in the later stage are still needed in order to identify potential ways to improve photosynthetic efficiency under salt stress.

## 4. Materials and Methods

### 4.1. Plant Materials and Treatments

In early October 2018, seeds of *C. paliurus* (Jinzhongshan No.11 family) were collected from Jinzhongshan county, Guangdong Province, China (24°58′ N latitude, 110°09′ E longitude). To break seed dormancy and promote prompt germination, seeds were treated with exogenous GA3 (gibberellin A3) and stratification measures [67]. In April 2019, the germinated seeds were sown in nonwoven containers [26]. After three months, uniform size seedlings (height: 40 ± 2.79 cm) were selected and transplanted to polypropylene containers (50L) with 1/2-strength Hoagland’s nutrient solution (pH 6.0 ± 0.2) that was continuously aerated with an air pump and renewed every seven days.

Two weeks after hydroponic transplanting, a salt stress experiment was carried in the Baima Experimental Base of Nanjing Forestry University (Nanjing, China) greenhouse [26]. Based on the previous study by Yao et al. [23], four salt concentration regimes were implemented: control (CK, 0 mM NaCl), 0.15% (LS, 25.7 mM), 0.30% (MS, 51.3 mM NaCl), and 0.45% (HS, 77.0 mM NaCl). Treatments were arranged in a complete randomized design with three biological replicates for each treatment, and each replication consisted of 8 seedlings with a seedling height of 45–55 cm and ground diameter of 3.8–4.2 mm.

### 4.2. Photosynthetic and Fluorescence Parameters Determination

In August 2019, determination of photosynthetic parameters and collection of leaf samples were carried out at 15 (T_1_) and 30 (T_2_) days after salt stress treatments. In each treatment, three seedlings were randomly selected and labeled as sample plants, and two fully expanded mature leaves were selected from the middle position of each selected plant for photosynthesis determination. Photosynthesis parameters including net photosynthetic rate (*P_n_*), transpiration rate (*T_r_*), stomatal conductance (*G_s_*), and intercellular CO_2_ concentration (*C_i_*) were measured by the LI-6400XT photosynthetic system (LI-COR, Inc., Lincoln, NE, USA) between 8:30–11:30 am. During determination, the gas flow rate was maintained at 200 μmol·s^−1^, leaf chamber temperature at 25 ± 0.5 °C, relative humidity at 60%, CO2 concentration at 400 ± 20 μmol·mol^−1^, and the photosynthetically active radiations (PAR) was set at 1200 μmol·m^−2^·s^−1^.

Fluorescence parameters of the selected leaves were determined by an FMS-2 portable pulse-modulated fluorometer (Hansatech Instruments Ltd., Norfolk, UK). After 20 minutes of dark adaptation, the selected leaves were used for the determination of maximal quantum yield of PS II (F_v_/F_m_), non-photochemical quenching coefficient (NPQ), photochemical quenching coefficient (qP), and electron transfer rate (ETR).

After determining photosynthetic and fluorescence parameters, fresh leaves were collected from the same part of the seedlings, washed and dried, and extracted with 80% acetone solution to determine the chlorophyll content following the method of Asghar et al. [68].

### 4.3. Carbon and Nitrogen Measurement

At each sampling time, one sample seedling of *C. paliurus* was selected for each replication based on the seedling height and ground diameter, and four intact and healthy leaves were sampled from each position of the upper, middle and lower, respectively. In total, 12 leaves were collected for each sample plant, in which half of the leaves were quickly frozen with liquid nitrogen, then stored at −80 °C for metabolites and total RNA extraction. The remaining leaves were dried at 70 °C for 72 h, then crushed, and sieved through 100 meshes, provided the material for leaves carbon and nitrogen contents determination using 50 mg of dry leaf powder, packed in foil, then placed in an element analyzer (vario MACRO cube, Elementar, Heraeus, Germany).

### 4.4. Metabolite Extraction and LC-MS/MS Analysis

Metabolite extraction and LC-MS/MS analysis for leaf samples were performed as previously described by Zhang et al. [26], using a UHPLC system (1290, Agilent Technologies, Santa Clara, CA, USA), coupled with a UPLC HSS T3 column (2.1 mm 100 mm, 1.8 μm) and Q Exactive (Orbitrap MS, Thermo, Waltham, MA, USA). The mobile phase A was 0.1% formic acid in water for positive, and 5 mmol/L ammonium acetate in water for negative, and the mobile phase B was acetonitrile. The elution gradient was set as follows: 0 min, 1% B; 1 min, 1% B; 8 min, 99% B; 10 min, 99% B; 10.1 min, 1% B; 12 min, 1% B. The flow rate was 0.5 mL/min. OSI-SMMS (version 1.0, Dalian Chem Data Solution Information Technology Co., Ltd., Dalian, China) was used for peak annotation after data processing with an in-house MS/MS database. The metabolites were mapped to the Kyoto Encyclopedia of Genes and Genomics (KEGG) metabolic pathways. Pathway enrichment analysis was used to identify the difference in abundance of metabolites in central metabolic pathways, including glycolysis (ko00010), citrate cycle (ko00020), pentose phosphate pathway (ko00030), and carbon fixation in photosynthetic organisms (ko00710). The hierarchical cluster analysis of the identified metabolites was performed by TBtools software (Version 1.066, South China Agricultural University, Guangzhou, China) [69].

### 4.5. Transcriptome Data Acquisition and Analysis

Total RNA extraction, cDNA library construction, and sequencing for each sample were performed as described by Zhang et al. [26]. In brief, total RNA was extracted using Trizol reagent kit (Invitrogen, Carlsbad, CA, USA) and the quality was assessed using an Agilent 2100 Bioanalyzer (Agilent Technologies, Palo Alto, CA, USA). The cDNA fragments were purified with a QiaQuick PCR extraction kit (Qiagen, Venlo, The Netherlands). A total of 24 cDNA libraries were constructed for leaf samples (at two collection times in four different salt stress treatments, three biological replications) using Illumina HiSeqTM 4000. The raw reads from the transcriptome sequencing were filtered by Fastp (Version 0.18.0, HaploX Biotechnology, Shenzhen, China) and the mapped reads of each sample were assembled using StringTie v1.3.1 (Center for Computational Biology, Johns Hopkins University, Baltimore, MD, USA) [70]. KEGG pathway enrichment analysis was adopted to evaluate metabolic pathways and related gene functions.

By means of the OmicShare tools (https://www.omicsmart.com/, accessed on 1 November 2021), a weighted gene co-expression network analysis (threshold power = 8, merge cut height = 0.75, and min module size = 50) was constructed to explore molecular regulatory mechanisms involved in photosynthesis. Modules with the significant enrichment of differential expression of genes (DEGs) in the photosynthesis pathway were selected, and the 2-kb promoter sequences of these DEGs were assessed for co-regulated cis-acting elements using MEME Suite software (Version 4.9.0, Department of Computer Science and Engineering, University of California at San Diego, San Diego, CA, USA) [71], which were used to identify transcription factors that would likely target these genes. The expression profiles for the photosynthetic and TF genes were obtained from transcriptome data. A co-expression analysis based on Pearson’s product-moment correlation (*p* <0.05, |r| > 0.75) was then performed to detect gene expression correlation networks. The gene regulatory networks were constructed by Cytoscape software (Version 3.7.1, Institute for Systems Biology, Seattle, WA, USA) [72].

### 4.6. Statistical Analysis

The statistical analysis of the data was computed using IBM SPSS Statistics 19 software (International Business Machines Corporation, New York, NY, USA) at a 95% confidence interval and *p* < 0.05 was considered significant.

## 5. Conclusions

In summary, the photosynthetic efficiency, chlorophyll content, and photochemical conversion efficiency in the leaves of *C. paliurus* seedlings were significantly affected by salt stress. By increasing the salt concentrations and prolonging the treatment periods, the photosynthetic system was more severely impaired by salinity. Salt stress not only significantly affected photosynthetic characteristics of *C. paliurus* seedlings, but also severely modified the abundance of metabolites (such as fumaric acid, sedoheptulose-7-phosphate, d-fructose-1,6-bisphosphate, and 3-phospho-d-glyceroyl phosphate) involved in glycolysis, pentose phosphate pathway, and tricarboxylic acid cycle, as well as the expression of photosynthetic genes. Based on the co-expression network analysis, 27 transcription factors (including *ERF*s, *IDD*, *DOF*, *MYB*, *RAP*) were identified to regulate photosynthetic genes under salt stress. Overall, this is the first report to expound the response mechanism in the photosynthetic system of *C. paliurus* subjected to salinity from the perspective of metabolism and transcription. Our findings could contribute to a deeper understanding of the response mechanism of *C. paliurus* seedlings to salinity and would provide a possibility to improve the photosynthetic capacity of *C. paliurus* through genetic engineering techniques.

## Figures and Tables

**Figure 1 ijms-23-01161-f001:**
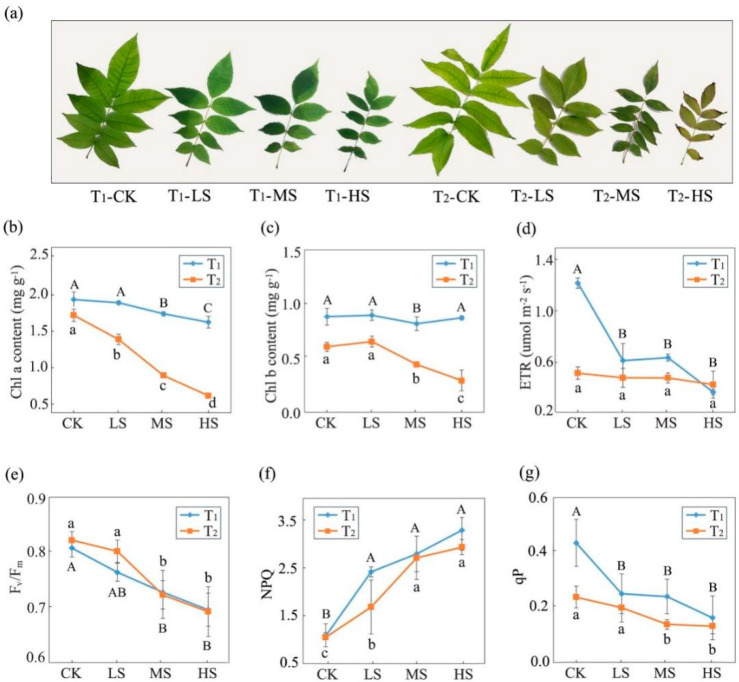
*C. paliurus* seedlings under different salt treatments at two sampling times. (**a**) Phenotypes, (**b**,**c**) chlorophyll content, and (**d**–**g**) fluorescence characteristic. Capital and lower-case letters represent significant differences among treatments at T_1_ and T_2_, respectively. T_1_, T_2_ represent two sampling times (15 and 30 days after treatments). CK, LS, MS and HS represent four NaCl treatment levels (control, 0.15%, 0.30% and 0.45%, *m/v*).

**Figure 2 ijms-23-01161-f002:**
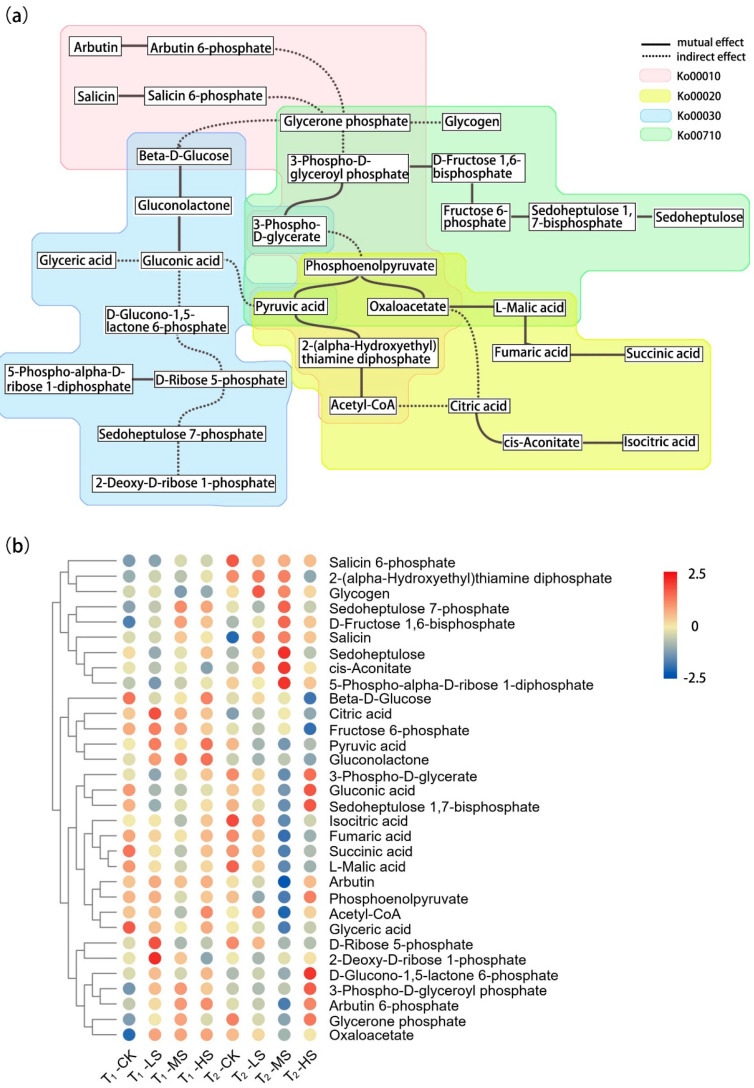
Changes in metabolite levels of *C. paliurus* leaves under different salt treatments and two sampling times. (**a**) Schematic diagram of metabolic pathways related to central metabolic processes (ko00010, glycolysis; ko00020, citrate cycle; ko00030, pentose phosphate pathway; ko00710, carbon fixation in photosynthetic organisms), and (**b**) hierarchical clustering heatmap of metabolites abundance (circle colors from red to blue represent metabolites abundance from high to low) (see Figure 1 for sampling time and treatments code).

**Figure 3 ijms-23-01161-f003:**
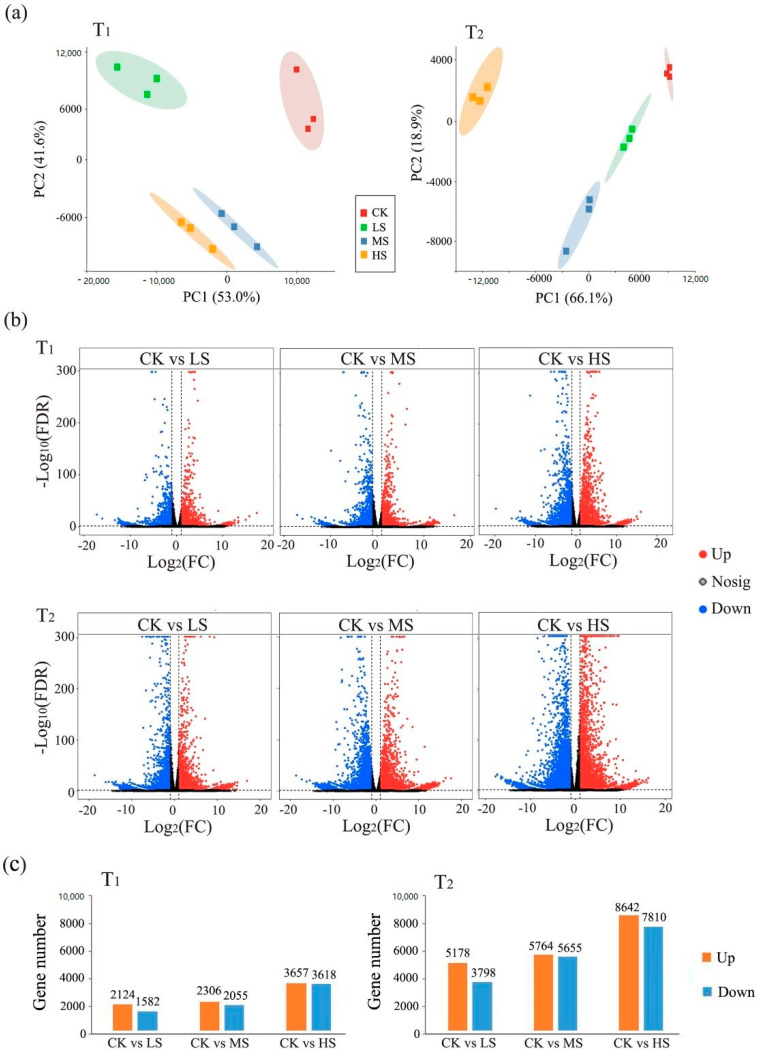
PCA score plots of gene expression, volcano plots of differentially expressed genes (DEGs), and the number of DEGs between different treatments at each sampling time. (**a**) Each point in the PCA score plot represents an independent biological replication, where red, green, blue, and yellow indicate CK, LS, MS, and HS, respectively; (**b**) Volcano plot of the DEGs (|Log2FC| > 1 and FDR < 0.05) between CK and the different salt treatments; (**c**) The numbers of DEGs between samples from different treatments (see Figure 1 for sampling time and treatments code).

**Figure 4 ijms-23-01161-f004:**
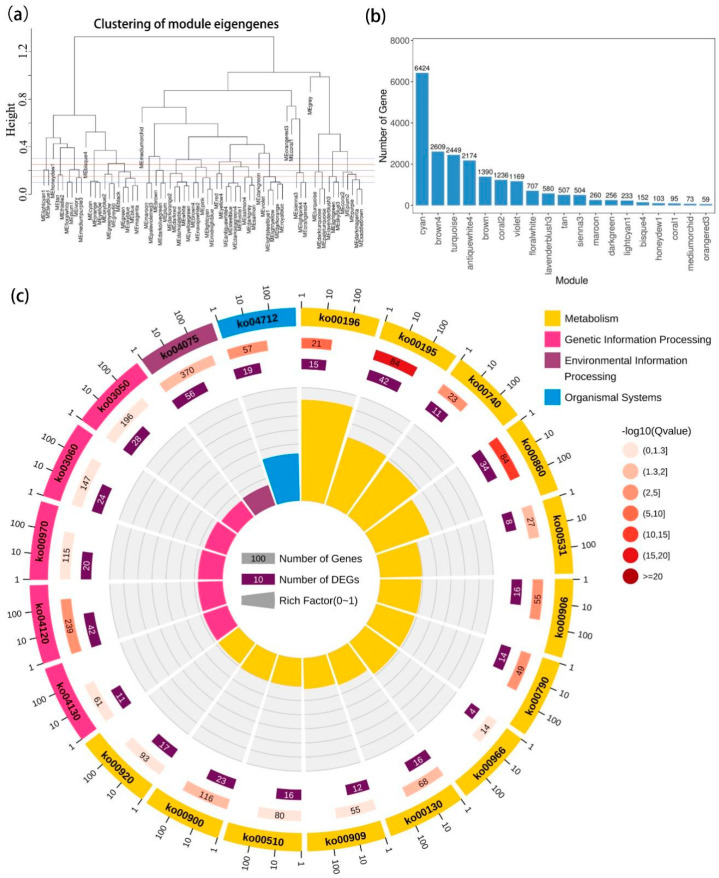
Gene co-expression network analysis. (**a**) Dendrogram showing modules identified by the weight gene co-expression network analysis and clustering dendrogram of module eigengenes; (**b**) The number of genes in each module; (**c**) KO (KEGG Ontology) enrichment circle diagram of cyan module (from the outside to the inside, the first circle represents the top 20 enrichment pathways, and the number outside the circle is the coordinate ruler of the number of genes; The second circle represents the number and Q value of background genes in this pathway, and the more genes, the longer the bar; The third circle represents the number of the DEGs in this pathway; The fourth circle represents the value of Rich Factor in each pathway) (Rich Factor refers to the ratio of DEGs to background genes).

**Figure 5 ijms-23-01161-f005:**
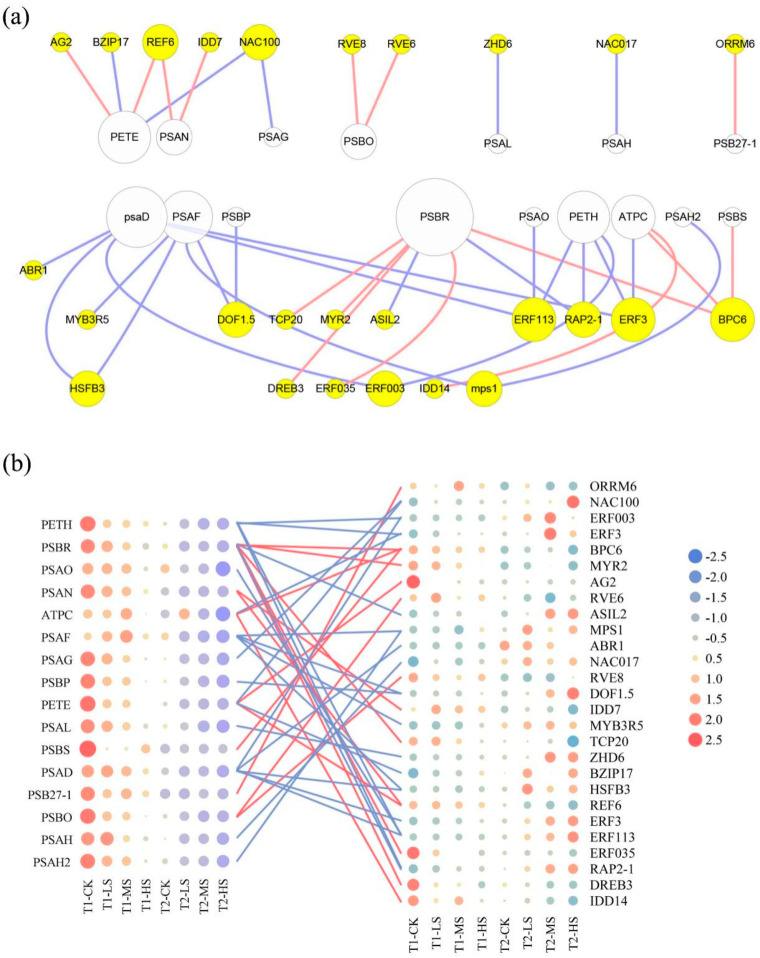
Co-expression relationship between TFs and photosynthesis genes. (**a**) Network constructed by selected TFs (yellow circles) and photosynthesis genes (white circles), and the size of the circles is directly proportional to the connectivity of genes; (**b**) Heatmap of photosynthesis genes (on the left) and TFs expression (on the right); The circle color from red to blue represents gene expression from high to low; red and blue lines represent positive and negative correlations between genes, respectively.

**Figure 6 ijms-23-01161-f006:**
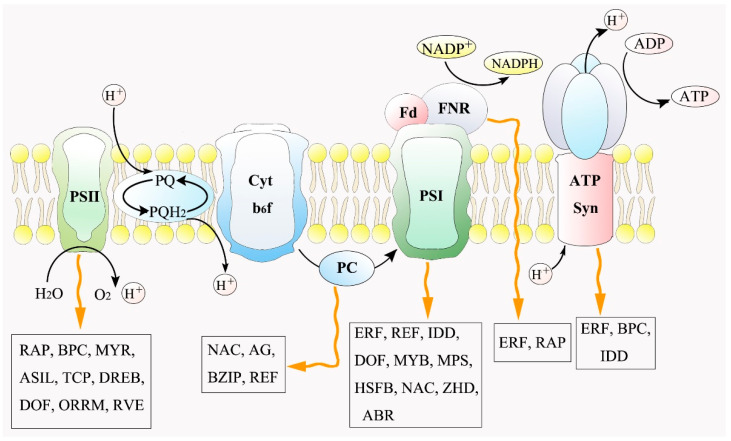
Schematic diagram of TFs involved in regulating photosynthetic process under salt stress. (TFs were screened based on co-expression network analysis).

**Table 1 ijms-23-01161-t001:** *C. paliurus* seedlings photosynthetic characteristics under different salt treatment times and concentrations.

Sampling Time	Treatment	Photosynthetic Parameters			
		*C_i_* (μmol·mol^−1^)	*T_r_* (mmol·m^−2^·s^−1^)	*G_s_* (mol·m^−2^·s^−1^)	*P_n_* (μmol·m^−2^·s^−1^)
T_1_	CK	312.67 ± 24.01 c	2.03 ± 0.15 a	80.67 ± 1.53 a	12.13 ± 1.10 Aa
	LS	342.00 ± 6.93 b	1.70 ± 0.10 b	72.00 ± 4.58 b	8.77 ± 129 Ab
	MS	351.33 ± 11.37 b	1.50 ± 0.09 bc	69.67 ± 2.08 b	5.27 ± 1.27 Ac
	HS	388.33 ± 07.02 a	1.40 ± 0.12 c	48.67 ± 5.03 c	3.77 ± 0.89 Ac
T_2_	CK	273.33 ± 08.15 c	2.07 ± 0.26 a	83.00 ± 4.58 a	9.83 ± 0.06 Ba
	LS	342.67 ± 23.59 b	1.43 ± 0.15 b	47.33 ± 4.04 b	4.40 ± 0.20 Bb
	MS	378.33 ± 05.69 a	1.23 ± 0.12 b	37.67 ± 5.03 c	2.93 ± 0.06 Bc
	HS	387.33 ± 03.06 a	1.10 ± 0.20 b	35.33 ± 3.05 c	2.63 ± 0.06 Bd

Different capital letters indicate a significant difference (*p* < 0.05) between two sampling times under the same salt concentration for *P_n_*. And different small letters indicate significant differences (*p* < 0.05) among different treatments at each sampling time (see Figure 1 for sampling time and treatments code).

## Data Availability

The Illumina raw sequencing profiles were submitted to the NCBI BioProject data-base under number PRJNA700136.

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
