# Peer review of "Metabolome and Transcriptome Analyses Unravel the Molecular Regulatory Mechanisms Involved in Photosynthesis of Cyclocarya paliurus under Salt Stress"

_ijms, 2022, doi:10.3390/ijms23031161_

Round 1
Reviewer 1 Report
The manuscript entitled “Metabolome and transcriptome analyses unravel the molecular regulatory mechanisms involved in photosynthesis of Cyclocarya paliurus under salt stress”, authored by Lei Zhang, Zijie Zhang, Shengzuo Fang, Yang Liu, and Xulan Shang, deals with the investigation of the photosynthetic characteristics of C. paliurus seedlings under different salt concentrations. Moreover, metabolome and transcriptome analyses were conducted by the authors in order to unravel the molecular regulatory mechanisms involved under the observed biological evidences. Only minor modifications are required before consider the manuscript suitable for the publication in International Journal of Molecular Sciences.
The revised manuscript is written with authority, and it contains very interesting data. The introduction summarizes the current state of the art, and perfectly introduces the reasons why the authors decided to start this study. However, I think that the materials and methods section is not well described. In particular, the punctual methodologies should be more descriptive, and this section should contain all the information necessary to replicate the experiments. For example, in section 4.4., The analysis parameters in mass spectrometry are not reported. The column used for separation either. The gradient and the solvents themselves are not shown. This problem is present more or less throughout this section. Consequently, I strongly suggest the authors to fix the problem related to this section adding more detail before submitting the manuscript again.
Keywords should be words not contained in the title or in the abstract. Their usefulness is to make easier the searching of the article using the common scientific search engines. Since several keywords are already present in the title, and/or repeated several times in the abstract, I strongly advise the authors to replace some of them and add more. As journal guidelines clearly report, a limited number of keywords can be used. Consequently, authors should carefully choose them.
Another problem with the revised manuscript is related to the use of not very recent bibliographic references. Again, I would suggest that authors introduce articles published in the last five years as references.
Author Response
Reviewer #1:
- However, I think that the materials and methods section is not well described. In particular, the punctual methodologies should be more descriptive, and this section should contain all the information necessary to replicate the experiments. For example, in section 4.4., The analysis parameters in mass spectrometry are not reported. The column used for separation either. The gradient and the solvents themselves are not shown. This problem is present more or less throughout this section. Consequently, I strongly suggest the authors to fix the problem related to this section adding more detail before submitting the manuscript again.
Response: Yes, you are correct, we have added detailed information about the experiment. For the detail, please see lines 387-390 in this revised version.
- Keywords should be words not contained in the title or in the abstract. Their usefulness is to make easier the searching of the article using the common scientific search engines. Since several keywords are already present in the title, and/or repeated several times in the abstract, I strongly advise the authors to replace some of them and add more. As journal guidelines clearly report, a limited number of keywords can be used. Consequently, authors should carefully choose them.
Response: According to the suggestion, we have modified the section of keywords. For the detail, please see lines 27-28 in this revised version.
- Another problem with the revised manuscript is related to the use of not very recent bibliographic references. Again, I would suggest that authors introduce articles published in the last five years as references.
Response: As recommended, we have introduced articles published in the last five years as references. For the detail, please see lines 468-476 in this revised version.
Reviewer 2 Report
Authors have investigated the transcript reprograming in leaves of Cyclocarya paliurus seedlings as related to the impairment of photosynthetic activity under increasing salt stress. The investigation is properly designed, and the interpretation of the results is correct.
Several investigations with a variety of plants have been previously published, reporting similar inhibitory effect of salt on photosynthesis and changes in levels of specific metabolites. To look for regulatory steps affected by salt stress, the manuscript describes the effect of increasing salt on transcriptome profile where several genes for proteins of photosynthetic machinery, stress response and transcriptional factors are identified.
The results have interest, but I would like to see a deeper discussion in comparison with similar investigations in other plants. In addition, I consider that accompanying measurements of enzyme activities would add scientific relevance in these investigations. At least, authors should reference and discuss precedent enzyme activity investigations. A final question, what about the relevance of the reported results in seedling for adult, full developed, plants?
There are also minor remarks to attend.
Line 67. Change “which is” to “that are”.
Line 70. Change “duo” to “due”.
Repeat in the legend for Figure 1 (part a, line 145) the meaning of colour of metabolic ways as indicated in line 382.
Author Response
Reviewer #2:
- The results have interest, but I would like to see a deeper discussion in comparison with similar investigations in other plants.
Response: As recommended, we have added a short discussion in comparison with similar investigations in other plants. For the detail, please see lines 223-229 in this revised version.
- In addition, I consider that accompanying measurements of enzyme activities would add scientific relevance in these investigations. At least, authors should reference and discuss precedent enzyme activity investigations.
Response: As recommended, we have referenced precedent enzyme activity investigations. For the detail, please see lines 231-232 in this revised version.
- A final question, what about the relevance of the reported results in seedling for adult, full developed, plants?
Response: Thanks for your comments. For the same plant or tree species, we speculate that the regulation mechanism of adult trees in response to salt stress is generally consistent with the results of this study, but the adult trees could be more tolerant to salinity. Unfortunately, we did this research only based on the seedling material under green house condition. and we will include your comment in the next filed experiment for further research. Thank you very much.
- Line 67. Change “which is” to “that are”.
Response: Thank you, we have corrected. For the detail, please see lines 66-67 in this revised version.
- Line 70. Change “duo” to “due”.
Response: Corrected (see line 72).
- Repeat in the legend for Figure 2 (part a, line 145) the meaning of colour of metabolic ways as indicated in line 382.
Response: As recommended, we have expounded the meaning of color of metabolic ways in the legend for Figure 2a. For the detail, please see lines 147-148 in this revised version.